# Circulating microRNAs as Potential Diagnostic Tools for Asthma and for Indicating Severe Asthma Risk

**DOI:** 10.3390/ijms26146676

**Published:** 2025-07-11

**Authors:** Elena V. Vorobeva, M. Aref Kyyaly, Collin L. Sones, Peijun J. W. He, S. Hasan Arshad, Tilman Sanchez-Elsner, Ramesh J. Kurukulaaratchy

**Affiliations:** 1Clinical and Experimental Sciences, Faculty of Medicine, University of Southampton, Southampton SO16 6YD, UK; e.v.vorobeva@soton.ac.uk (E.V.V.); sha@soton.ac.uk (S.H.A.); t.sanchez-elsner@soton.ac.uk (T.S.-E.); 2Biomedical Science, Department of Science and Engineering, Solent University Southampton, Southampton SO14 0YN, UK; aref.kyyaly@solent.ac.uk; 3Optoelectronics Research Centre, Faculty of Engineering and Physical Sciences, University of Southampton, Southampton SO17 1BJ, UK; cls@orc.soton.ac.uk (C.L.S.); phe@highfielddiagnostics.co.uk (P.J.W.H.); 4Highfield Diagnostics Ltd., Southampton SO17 1PJ, UK; 5NIHR Southampton Biomedical Research Centre, University Hospitals Southampton, Southampton SO16 6YD, UK; 6The David Hide Asthma and Allergy Research Centre, NewPort PO30 5TG, UK

**Keywords:** asthma severity, biologic, biomarker, diagnosis, microRNA

## Abstract

Asthma places a significant burden at individual and societal levels, but there remains no gold-standard objective test for asthma diagnosis or asthma severity risk prediction. MicroRNAs (miRNAs) are short non-coding RNA sequences that are attracting interest as biological signatures of health and disease status. We sought to construct serum miRNA panels that could serve as potential biomarkers to aid in the diagnosis of asthma and predict asthma severity. Thirty-five asthma-related miRNAs were screened in the serum of three patient groups (never-asthma, mild-asthma, and severe-asthma; *n* = 50/group) drawn from two well-characterised cohorts. miRCURY LNA technology was used, followed by GeneGlobe analysis. The associations of miRNA expression with clinical outcomes of interest and diagnostic value of the proposed miRNA panels were assessed. We identified an asthma diagnosis panel comprising upregulated miR-223-3p, miR-191-5p, and miR-197-3p (area under curve (AUC) = 0.813, sensitivity 76% and specificity 72%). Compared with mild-asthma individuals, we also identified an asthma severity risk panel comprising upregulated miR-223-3p plus downregulated miR-30a-5p, miR-660-5p, and miR-125b-5p (AUC = 0.759, sensitivity 78%, specificity 64%). Individual miRNAs showed associations with worse clinical asthma severity and impaired quality of life. miRNA panels with high sensitivity and specificity offer potential as biomarkers for asthma diagnosis and asthma severity.

## 1. Introduction

Asthma is a common chronic respiratory condition affecting over 300 million people worldwide [1]. Though the availability of effective asthma treatments has expanded considerably, especially for more complex disease, asthma continues to impose a significant burden at both societal and individual levels [2]. The diagnosis of asthma remains clinical, based on the assessment of multiple clinical parameters, with an ongoing absence of a gold-standard objective diagnostic test. That leaves asthma diagnosis open to subjectivity, complexity and uncertainty, which collectively may lead to underdiagnosis, delayed diagnosis, and delayed treatment [3]. Asthma severity is similarly unsatisfactorily defined by the level of treatment required to control disease where disease control is subjectively defined [4,5,6]. A composite measure of asthma severity incorporating asthma treatment level, asthma control, lung function impairment, exacerbation frequency, and need for higher-level asthma treatments has been proposed (asthma severity scoring system (ASSESS) score) [7]. However, such classifications mean that the disease has to become a problem before its severity is recognised and appropriate treatment is applied. Early, accurate identification of asthma patients at risk of severe disease could guide strategies to improve their management at earlier stages and potentially avert downstream adverse outcomes. The development of reliable objective markers to support precise asthma diagnosis and promptly differentiate severe from mild asthma is a pressing need. 

MicroRNAs (miRNAs) offer one potential biomarker avenue. They are small non-coding single-stranded RNAs (ribonucleic acids), 22–25 nucleotides long, acting via RNA-induced silencing complexes to post-transcriptionally regulate mRNAs (messenger RNA) possessing complementary sequences [8]. Highly stable circulating miRNAs occur in biological fluids, including peripheral blood, and are potential biomarkers for diagnosis, prognosis, and disease monitoring [9,10,11]. Growing evidence indicates that miRNAs are differentially expressed in asthmatics compared with never-asthmatics and have immunoregulatory effects [12,13,14,15,16]. In a recent systematic review, we identified a panel of candidate miRNAs which were differentially expressed in the sera of patients with asthma compared with those without (upregulated in asthma: miR-155, miR-126, miR-125b, miR-21, miR-98, and miR-146a, downregulated in asthma: miR-192, miR-15a, Let-7, and miR-30a) [17]. The review also highlighted miRNAs that were differentially expressed in severe compared with mild asthma (upregulated in severe asthma: miR-155, miR-126, miR-125b, and miR-1165-3p, downregulated in severe asthma: miR-1 and miR-19b) and that thus might be potential biomarkers of asthma severity risk [17]. This concept has been further reinforced by recent reports of differential miRNA expression in severe compared with mild asthma [18,19,20]. Therefore, further investigation of miRNAs as supportive tools for asthma diagnosis and to identify severity risk is warranted.

In this study, we screened miRNA in sera from three groups of well-characterised participants in two cohort studies who were classified as (a) never-asthma, (b) mild-asthma, or (c) severe-asthma. Our aim was to identify miRNAs in serum that could serve as potential biomarkers to (i) aid in the diagnosis of asthma compared with not having asthma, and (ii) distinguish mild from severe asthma, to construct potential diagnostic panels using the identified miRNAs.

## 2. Results

### 2.1. Demographics

Table 1 presents the demographic characteristics of three groups: never-asthma, mild-asthma, and severe-asthma. The severe-asthma group had a higher female predominance (70% in the severe-asthma group compared with 60% and 54% in the mild-asthma and never-asthma groups, respectively), and a significantly higher body mass index (BMI) (28.9 for the severe-asthma group compared with 25.3 and 25.1 for the mild-asthma and never-asthma groups, respectively, *p* < 0.001). Asthma onset was also significantly younger in mild cases (*p* < 0.001). Atopy was most common in the mild-asthma group, while smoking rates were similar across all groups.

### 2.2. miRNA for Asthma Diagnosis (GeneGlobe Data and Receiver Operating Characteristic (ROC) Curve Analysis)

First, we compared never-asthma participants (*n* = 50) with all asthmatic individuals (combining mild asthma and severe asthma, *n* = 100) (Figure 1a). The analysis revealed that, out of the 35 microRNAs screened, 3 microRNAs were differentially expressed in the sera of asthma patients in comparison with never-asthma individuals: miR-223-3p, miR-191-5p, and miR-197-3p were upregulated 1.38- (*p* < 0.0001), 1.44- (*p* < 0.0001), and 1.33- (*p* < 0.0001) fold, respectively, in the asthma group (Figure 1b; Appendix A).

The diagnostic values of these miRNAs were evaluated by ROC curve analysis. The ROC area under curve (AUC) and 95% confidence interval (CI) for individual asthma diagnosis miRNAs were as follows: 0.700 (0.610–0.791) for miR-223-3p, 0.799 (0.726–0.871) for miR-191-5p, and 0.715 (0.621–0.809) for miR-197-3p (Figure 1c; Appendix A). To increase diagnostic accuracy, multiple logistic regression combining all diagnostic miRNAs was carried out and evaluated by ROC curve analysis (Appendix A). The resulting combination revealed better performance with an AUC value of 0.813 (0.743–0.883), sensitivity of 76%, and specificity of 72%. 

### 2.3. miRNA for Asthma Severity and Associated Clinical Parameters

The comparison of mild-asthma with never-asthma individuals showed only 2 miRNAs, miR-191-5p and miR-197-3p, that were significantly upregulated 1.3- (*p* < 0.0001) and 1.28- (*p* < 0.001) fold, respectively, in mild-asthma samples (Figure 2a; Appendix A). Both miRNAs were also observed to be upregulated in the severe-asthma group, with significantly higher upregulation in severe-asthma in comparison with mild-asthma individuals in the case of miR-191-5p (1.23-fold, *p* < 0.0001).

The comparison of severe-asthma with mild-asthma individuals also revealed one upregulated (miR-223-3p, 1.37-fold, *p* < 0.0001) and three downregulated miRNAs (miR-30a-5p, 1.4-fold, *p* < 0.001; miR-660-5p, 1.25-fold, *p* < 0.01; miR-125b-5p, 1.31-fold, *p* < 0.05) in severe-asthma patients (Figure 2b; Appendix A). Interestingly, those miRNAs were unchanged in the mild-asthma group in comparison with never-asthma individuals, which makes them unique to severe-asthma status. 

Understanding the association of miRNA expression with different clinical parameters of asthma severity might also be informative in the context of asthma severity risk identification.

To determine the connection of miRNAs with asthma severity characteristics, we compared the levels of expression between the top and bottom quartiles of a variable, as well as between subgroups stratified by conventional cut-off values for the clinical parameter of interest. To assess whether sera microRNAs differentially expressed in asthmatic individuals could be linked to lung function, we evaluated the association of screened miRNAs with forced expiratory volume in one second (FEV1) between asthma groups. The comparison of miRNAs expression between FEV1 bottom and top quartiles among asthma patients showed that, for miR-30a-5p, miR-155-5p, and miR-15a-5p, downregulation was associated with diminished lung function (lower FEV1 quartile) (Figure 3a, Figure 3b, and Figure 3c, respectively).

We also created a modified asthma severity score (m-ASSESS) substituting the Asthma Control Questionnaire (ACQ-6) for the Asthma Control Test (ACT) within this multidimensional score, given that ACQ-6 was the available measure of asthma control in the Wessex AsThma CoHort of difficult asthma (WATCH) study (see Appendix A). 

Upregulation of miR-223-3p and miR-197-3p was observed in the top quartile of m-ASSESS (Figure 3d,e). The quartile stratification of the samples is shown in Appendix A. miR-197-3p was also significantly upregulated in the group of patients with inadequate asthma control (ACQ-6 > 1.5) (Figure 3f).

The top quartile of fractional exhaled nitric oxide (FeNO) values showed higher expression of miR-191-5p (Figure 3g). Both miR-191-5p and miR-151a-3p were upregulated in patients with higher level of airway inflammation (FeNO > 40 ppb, Figure 3h,i). miR-191-5p was also more abundant in the sera of severe-asthma patients on maintenance (daily) oral corticosteroid (m-OCS) (Figure 3j).

Finally, the association of miRNA expression with quality of life (St. George’s Respiratory Questionnaire, SGRQ) of severe-asthma patients showed downregulation of miR-15a-5p in the top (worst) quartile of SGRQ score (Figure 3k).

Thus, we showed that several miRNAs are associated with various clinical parameters and could be potential markers of airway inflammation and asthma severity. The predictive values of these miRNAs for asthma severity were evaluated by ROC curve analysis. The ROC AUC and 95% CI were as follows: 0.724 (0.624–0.824) for miR-223-3p, 0.713 (0.607–0.819) for miR-30a-5p, 0.685 (0.579–0.791) for miR-660-5p, and 0.643 (0.534–0.752) for miR-125b-5p. When all four miRNAs were combined, the sensitivity was 78%, specificity was 64%, and AUC was 0.759 (0.663–0.855) (Figure 2c and Appendix A).

### 2.4. Targets and Biological Pathways of miRNAs Differentially Expressed in the Sera of Severe-Asthma Patients

To determine the potential target genes and associated biological pathways of miRNAs implicated in severe-asthma, miRSystem was used [21]. A list of miRNAs differentially expressed in the sera of severe-asthma patients in comparison with never-asthma individuals (Appendix A) was uploaded to miRSystem, which generated the target gene summary report and functional annotation summary report.

The enriched pathways identified (Figure 4a) were associated with mammalian carbohydrate interconversions including involvement in steroid inactivation, cell processes (meiosis, chromosome maintenance, regulation of retinoblastoma protein that plays a role in cell division cycle, and transcription of rRNA), and development and homeostasis of various cells and tissues through forkhead box A (FOXA) transcription factors. A considerable proportion of the identified pathways (~40%) were related to inflammation and immune responses. Amongst them were the formation of proteinaceous fibrillar deposits (amyloids) that could complicate conditions characterised by chronic inflammation, cytokine signalling, including IL-4 and interferon (IFN) α/β, as well as antigen presentation and processing involving the proteasome.

The heatmap of target genes in identified terms (Figure 4b) demonstrates that cytokine signalling and the antigen presentation and processing pathways were more enriched and targeted by the higher number of miRNAs of interest. The miRNAs that targeted the identified pathways are shown in Appendix A. Those pathways played crucial roles in airway inflammation (hallmark of asthma) and asthma exacerbation (hallmark of asthma severity), and are the targets for asthma treatment with biologics like Omalizumab and Mepolizumab. 

## 3. Discussion

Using GeneGlobe analysis, we identified miRNAs of diagnostic value differentially expressed in the sera of asthma patients (miR-223-3p, miR-191-5p, and miR-197-3p) compared with the never-asthma group. ROC curve analysis showed that a combination of these three miRNAs had high predictive value for asthma diagnosis, with an AUC of 0.81. Further, we identified four miRNAs unique to severe-asthma status (miR-223-3p, miR-30a-5p, miR-660-5p, and miR-125b-5p). A combination of these four severity-related miRNAs had the ability to discriminate mild from severe asthma with relatively high sensitivity and specificity (AUC = 0.76). This notion is supported by the fact that these miRNAs were associated with asthma characteristics that indicate more severe asthma, such as lower lung function and higher ACQ-6.

These are the first diagnostic panels to be constructed to distinguish asthma and asthma severity that demonstrate acceptable levels of diagnostic accuracy, as evidenced by favourable AUC values and reasonable sensitivity, though comparatively lower specificity. Such test properties will be reliable for detecting the condition and ruling it out when the test is negative. In the context of asthma and prediction of its severity, being able to confidently ‘rule out’ the condition in question provides a potentially clinically useful tool to aid clinicians in their asthma diagnosis and management pathways, especially in situations of clinical doubt. This could potentially avoid inappropriate treatment implementation and prompt a search for alternative diagnoses. Future research could assess a wider miRNA array to pursue an improved specificity for this testing approach in asthma that has a better ability to ‘rule in’ the states under question. The incorporation of clinical parameters alongside miRNA panels may also, in the future, yield a diagnostic approach with better combined sensitivity and specificity.

Some of the identified miRNAs have been previously studied in asthma and other conditions. For example, miR-223-3p is easily detectable and has been reported to be upregulated in various tissues of asthma patients, like bronchial brushings [22], peripheral blood leukocytes [23], and sputum [24]. It is linked to higher eosinophils and neutrophils in tissue [16] and regulates immune cell proliferation, differentiation, and polarisation during inflammation [25]. Similarly, miR-155 was shown to be downregulated in asthmatic individuals compared with healthy subjects in exhaled breath condensates [12], asthmatic human bronchial epithelial cells [26], and nasal biopsy specimens [27]. Contrary to these studies, we did not find this miRNA to be downregulated in the serum of asthma patients. However, this miRNA was associated with diminished lung function in our study. Furthermore, some studies reported increased miR-155 levels in the serum [28] and plasma of asthma patients and their association with the degree of asthma severity [29,30]. One possible explanation for this discrepancy might be that we studied the specific miR-155-5p strand, whereas those previous studies reported findings for miR-155 at a broader level. However, both -3p and -5p miRNA strands might be functional and express functions that differ in one strand from another [31]. miR-155 is involved in the regulation of allergic inflammation [32] and suppresses Th2 immunity in allergic disorders [28]. Hence, it is conceivable that the downregulation of miR-155 promotes allergic inflammation. miR-191-5p and miR-197-3p were shown to be increased in the extracellular vesicles of asthma patients [33] and participate in eosinophilic inflammation and remodelling [34], respectively, indicating their role in asthma. In summary, these three miRNAs have been reported to be associated with asthma in various biological fluids or tissues. However, for the first time, we show that all three miRNAs can be detected in sera and are differentially expressed in asthma patients, and combining them can have diagnostic value in asthma.

The interest in miRNA expression association with the severity of asthma has evolved recently. We found four miRNAs that were differentially expressed in severe-asthma. Consistent with our severity-related miRNA findings, Maes et al. also demonstrated 223-3p to be significantly upregulated in the sputum of patients with severe asthma compared with that in healthy control subjects [24]. miR-660-5p has not been reported in asthma before, but was shown to be upregulated in lung fibroblasts of chronic obstructive pulmonary disease (COPD) patients [35]. miR-30a-3p was decreased in the peripheral blood [36] and was reported to attenuate fibrosis in asthma [37]. miR-125b was downregulated in the sputum [38], but increased in the plasma [13] and serum exosomes of asthma patients [39]. miR-125b-5p was downregulated in bronchial biopsies of asthmatic individuals [16]. Supportive to our findings, miR-125b has been linked to asthma severity [39]. We further explored the relationship between miRNA expression and various clinical parameters of asthma severity. Our findings show that the association of miR-30a-5p, miR-155-5p, and miR-15a-5p decreases with decreased lung function (lower FEV1 quartile) of asthma patients. Importantly, miR-30a-5p and miR-15a-5p were differentially expressed only in the severe-asthma group in comparison with never-asthma and mild-asthma individuals, which suggests that miRNAs may be relevant to both asthma development and severity. 

A further investigation of the association of miRNA expression in severity-associated clinical parameters in the severe-asthma group revealed the relation of increased levels of miR-223-3p and miR-197-3p to higher m-ASSESS scores, with miR-197-3p also upregulated in the sera of patients with inadequate asthma control. Upregulation of miR-151a-3p and miR-191-5p was observed in the sera of patients with greater airway inflammation and of individuals on m-OCS in the case of miR-191-5p, whereas downregulation of miR-15a-5p was associated with lower quality of life. These findings suggest the role of miRNAs under study as potential markers of airway inflammation and asthma severity.

To gain better insight into the role of miRNAs in the pathophysiology of asthma, an analysis of pathways and genes targeted by miRNA differentially expressed in the severe-asthma group was performed. Several asthma-related pathways were identified. One of them is the network of transcription factors, FOXA2–FOXA3, known to be inversely correlated with each other and contribute to mucous cell metaplasia in asthma [40]. Several pathways were associated with mammalian carbohydrate interconversions, including sucrose and starch metabolism, which were recently shown to be associated with an increased prevalence of asthma if consumed in excess [41]. 

The antigen presentation and cytokine signalling pathways, including IFNα/β and IL-4 signalling events, exhibited greater enrichment and were targeted by a higher number of miRNAs of interest. Those pathways play a vital role in airway inflammation in asthma. IFNα/β TLR9-mediated release by plasmacytoid dendritic cells is inhibited by FcεRI activation during viral airway infection and potentially increases the duration and/or severity of viral-induced airway damage [42]. Indeed, deficient type I IFN responses were observed in blood monocytes from patients with atopic asthma [43]. 

## 4. Materials and Methods

### 4.1. Study Design

The panel of miRNAs chosen for this work was based on our previous studies. First, a systematic review of studies reporting differential expression of specific miRNAs in the tissues and body fluids of adults and children with asthma across 3 databases (Medline, Embase, and SCOPUS) was conducted [17]. As described above, this review identified miRNAs that were reported in more than 2 publications to be differentially expressed in asthma patients compared with people without asthma. These miRNAs showed consistent expression in the blood of asthma patients, which enabled the construction of panels of miRNAs expressed in sera that can be used as potential non-invasive biomarkers for asthma diagnosis and severity risk assessment. Then, a pilot study assessing miRNAs expression in the sera from patients with mild-asthma, severe-asthma, and never-asthma individuals (N = 4 per group) was conducted using a miRCURY LNA miRNA Focus PCR Panel kit (QIAGEN, Hilden, Germany), which is designed for profiling 175 human miRNAs commonly found in serum and plasma [20]. This pilot study demonstrated miRNAs in sera that distinguished severe-asthma from both never-asthma and mild-asthma (Appendix A).

The asthma diagnosis and severity risk miRNA panels from the systematic review together with the miRNAs identified in our subsequent pilot study enabled us to design a panel of 35 miRNAs (Figure 1) whose expression was evaluated using miRCURY LNA miRNA custom PCR panels (QIAGEN, Hilden, Germany) in the sera collected from 3 groups of adults (N = 50 each). These groups were classified as severe asthma, never asthma, and mild asthma. 

### 4.2. Study Population

Participant samples and clinical data were obtained from 2 existing cohort studies. The Isle of Wight Whole Population Birth Cohort (IOWBC) was established in 1989 to prospectively study the natural history of asthma and allergies in participants (N = 1456) born on the Isle of Wight, United Kingdom (UK), between January 1st 1989 and February 28th 1990 [44]. Ethics approval was obtained for IOWBC from the Isle of Wight NHS Ethics Committee (No 05/89; dated 22 August 1988). The WATCH study (N = 500) was established in the Regional Difficult Asthma clinic at University Hospitals Southampton NHS Foundations Trust for patients who are managed with “additional controller medications” and/or “specialist therapies”, according to the British Thoracic Society (BTS) Adult Asthma Management Guidelines 2019 [45]. The study design and protocol were approved by the West Midlands–Solihull Research Ethics Committee (14/WM/1226). In both cohorts, participants gave written informed consent for their data and samples to be used in future ethically approved asthma research studies. The present study was approved by the Wales REC 6 (20/WA/0301).

For this study, never asthma was defined as IOWBC participants without a history of diagnosed asthma or recurrent wheezing at any timepoint during their life when assessed at 26 years. Mild asthma was defined as IOWBC participants with physician-diagnosed asthma at 26 years who were on Global Initiative for Management of Asthma (GINA) steps 1–2-equivalent treatment [6]. Severe asthma was defined as biologically naïve WATCH participants who met the ERS/ATS definition for severe asthma and were on GINA 2020 steps 4–5-equivalent treatment [4,6].

### 4.3. Sample Collection and RNA Extraction

For mild- and never-asthma participants, serum samples were collected at the 26-year IOWBC assessment and stored at –80 °C until analysis. For severe-asthma patients, serum samples were collected at WATCH enrolment and stored at –80 °C until analysis. miRNA from 200 uL of stored serum was extracted with a miRNeasy Serum/Plasma Advanced Kit (QIAGEN, Hilden, Germany) with the addition of spike-in controls (RNA Spike-in Kit, QIAGEN, Hilden, Germany) to provide controls for the quality of the RNA isolation.

### 4.4. qRT-PCR and Data Analysis

RNA was reverse-transcribed to cDNA using a miRCURY LNA RT Kit (QIAGEN, Hilden, Germany) with the addition of spike-in controls for quality control of cDNA synthesis using DNA Engine Tetrad 2 Cycler (Bio-Rad, Hercules, CA, USA). miRNA expression was evaluated by quantitative PCR on a 7900HT Fast Real-Time PCR System with a 384-well block module (Applied Biosystems, Foster City, CA, USA) using miRCURY LNA miRNA SYBR Green custom PCR Panels for Human Serum/Plasma (QIAGEN, Hilden, Germany) following the manufacturer’s protocol. The PCR data obtained were analysed using the QIAGEN web portal GeneGlobe (https://geneglobe.qiagen.com, accessed 19 June 2025), which provides a range of web-based tools for data analysis, including real-time PCR modules that transforms threshold cycle (Ct) values from QIAGEN PCR panels, to calculate results for miRNA expression applying the 2^−ΔΔCt^ method. The GeneGlobe settings were as follows: fold-change threshold > 1.2 and Ct ≤ 35. Data normalisation was performed using 6 reference miRs: miR-let-7i-5p, miR-148b-3p, miR-30e-5p, miR-222-3p, miR-425-5p, and miR-484. Haemolysis and spike-in controls for RNA extraction quality, RT efficiency, and PCR amplification plate-to-plate differences were also included.

### 4.5. Bioinformatics Analysis of Targets and Biological Pathways

The potential target genes and their associated biological pathways for miRNAs differentially expressed in the sera of severe-asthma patients compared with never-asthma individuals were determined with an online tool, miRSystem [21]. MiRSystem is an online tool that uses 7 well-known target prediction databases, experimentally validated data from TarBase and miRecords, and 5 functional annotation databases, and applies various statistical approaches (http://mirsystem.cgm.ntu.edu.tw, accessed 31 October 2023). The target gene summary report and functional annotation summary report were obtained with the following tool settings: 5 pathway databases (KEGG, Biocarta, PID, Reactome, GO tier 2), Hit ≥ 3, O/E ratio ≥ 2, and total genes in pathway ≥ 25 and ≤500. Pathways were chosen based on the empirical *p*-value cut-off of 0.05.

### 4.6. Clinical Parameters

Clinical outcomes of interest that were assessed in severe-asthma patients from WATCH included FEV1 and FeNO, and those for severe asthma also included ACQ-6, exacerbations needing oral corticosteroid (OCS) in the past 12 months, being on m-OCS, m-ASSESS score, and SGRQ. Associated methodology has been previously described for WATCH [45]. The details of ASSESS score calculation are given in Appendix A. 

### 4.7. Statistical Analysis

Descriptive statistics were used to summarise demographic characteristics across the three groups (severe-asthma, mild-asthma, never-asthma). Continuous variables were reported as medians with interquartile ranges and compared using the Kruskal–Wallis test due to the non-normal distribution. Categorical variables were reported as counts and percentages and compared using the Chi-square test. Statistical comparisons were only conducted for parameters available across all three groups. A *p*-value of <0.05 was considered statistically significant. All analyses were performed using IBM SPSS Statistics (version 29.2.0, Armonk, NY, USA). miRNA expression comparison between groups was performed using GeneGlobe with an unpaired 2-tailed *t*-test. The normality of the data was assessed with the Shapiro–Wilk test, followed by the Mann–Whitney U test and two-stage step-up method of Benjamini, Krieger, and Yekutieli false discovery rate, (Q) = 5%. The association of miRNA expression and clinical parameters, and statistical analyses were performed using GraphPad Prism software (version 10.1.2, Boston, MA, USA). The normality of the data was confirmed by the Shapiro–Wilk test. An unpaired 2-tailed *t*-test was applied, with a *p*-value of <0.05 considered statistically significant. For the diagnostic panel, multiple logistic regression combining predictors was carried out, followed by ROC curve analysis of the obtained probabilities using GraphPad Prism software (version 10.1.2). The optimal cut-off value, sensitivity, and specificity were determined by calculating the Youden index. ROC curve analysis was performed, and the ROC AUC with the 95% CI were calculated to evaluate the diagnostic value of individual miRNAs.

## 5. Conclusions

Thus, working with relatively large, well-defined groups of asthma patients in this study, we identified miRNAs that are differentially expressed in the sera of asthma patients and associated with several clinical parameters, which make them potent asthma diagnostic markers. We also defined a set of miRNAs that are unique to severe asthma, are associated with asthma severity indices, such as lower lung function and higher ACQ-6, and therefore might serve as asthma severity risk biomarkers. We identified specific pathways associated with those miRNAs, which helps to gain a better understanding of the role that miRNAs play in the complexity of asthma. Further studies in different populations and ethnicities are needed to validate these findings and determine their broader clinical relevance to progress our understanding of miRNA-based approaches in asthma diagnosis and management. 

## 6. Patents

E.V.V., M.A.K., S.H.A., T.S.-E., and R.J.K. are co-applicants on a UK patent filed in relation to the diagnostic and severity asthma miRNAs identified in this study (application number 2419126.4).

## Figures and Tables

**Figure 1 ijms-26-06676-f001:**
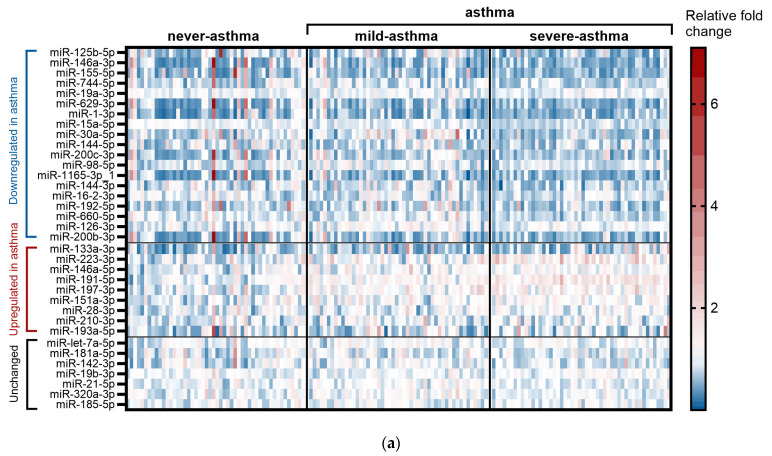
Circulating miRNAs differentially expressed in the sera of asthma patients and their diagnostic value. (**a**) Heat map of all miRNAs screened stratified by up/downregulation in the asthma group (includes mild-asthma, *n* = 50 and severe-asthma, *n* = 50) in comparison with never-asthma individuals (*n* = 50). Fold change was calculated using the mean of miRNA expression in the never-asthma group as a reference. (**b**) Relative expression of miRNAs (dCt; normalised to 6 reference miRNAs) in the asthma group (*n* = 100) in comparison with never-asthma individuals (*n* = 50). (**c**) Receiver operating characteristic (ROC) curves of individual asthma diagnosis miRNAs and their combined panel. AUC—area under the ROC curve; CI—confidence interval. miRNA expression was evaluated by qPCR using miRCURY LNA technology followed by GeneGlobe analysis (QIAGEN) with the following settings: fold-change threshold > 1.2, Ct ≤ 35. Statistical analysis: min to max with line at median, Mann–Whitney U test, **** *p* < 0.0001.

**Figure 2 ijms-26-06676-f002:**
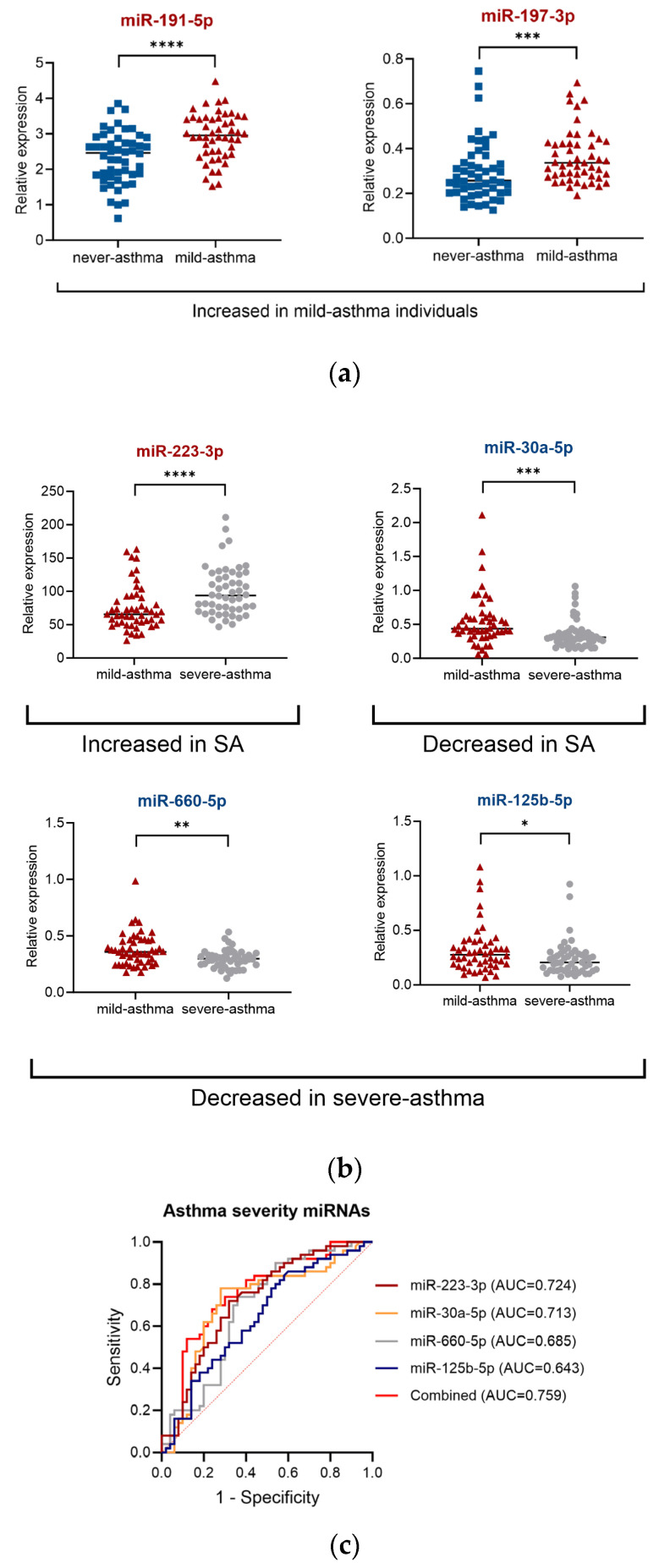
miRNAs differentially expressed in the sera of severe-asthma patients and their diagnostic value. (**a**) Mild-asthma patients (*n* = 50) in comparison with never-asthma individuals (*n* = 50), (**b**) severe-asthma group (*n* = 50) in comparison with mild-asthma cohort—severity of miRNAs, (**c**) receiver operating characteristic (ROC) curves of individual severity of miRNAs and their combined panel. AUC—area under the ROC curve; CI—confidence interval. miRNA expression evaluated by qPCR using miRCURY LNA technology followed by GeneGlobe analysis (QIAGEN); dCt (normalised to 6 reference miRNAs). GeneGlobe settings: fold-change threshold > 1.2, Ct ≤ 35, Mann–Whitney U test, * *p* < 0.05, ** *p* < 0.01, *** *p* < 0.001, **** *p* < 0.0001.

**Figure 3 ijms-26-06676-f003:**
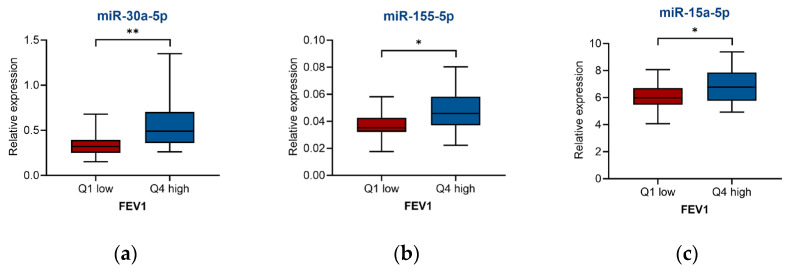
Association of miRNA expression in the sera of asthma patients with clinical parameters of asthma. miRNAs expression in (**a**–**c**) asthma group FEV1 bottom and top (*n* = 19) quartiles, and (**d**,**e**) severe-asthma group severity score (modified ASSESS) bottom and top quartiles. (*n* = 7), (**f**) severe-asthma group ACQ6 in groups divided by a conventional cut-off (ACQ6 < 1.5, *n* = 14; ACQ6 > 1.5, *n* = 34), (**g**–**i**) severe-asthma group FeNO bottom and top quartiles (*n* = 11) and in groups divided by a conventional cut-off (FeNO < 40, *n* = 34; FeNO > 40, *n* = 8), (**j**) severe-asthma groups of patients that were (*n* = 14)/were not (*n* = 36) on maintenance oral corticosteroids (m-OCS), (**k**) severe-asthma group quality of life score (SGRQ) bottom and top quartiles (*n* = 11). dCt, normalised to 6 reference miRNAs. Statistical analysis: min to max with line at median, unpaired 2-tail *t*-test, * *p* < 0.05, ** *p* < 0.01.

**Figure 4 ijms-26-06676-f004:**
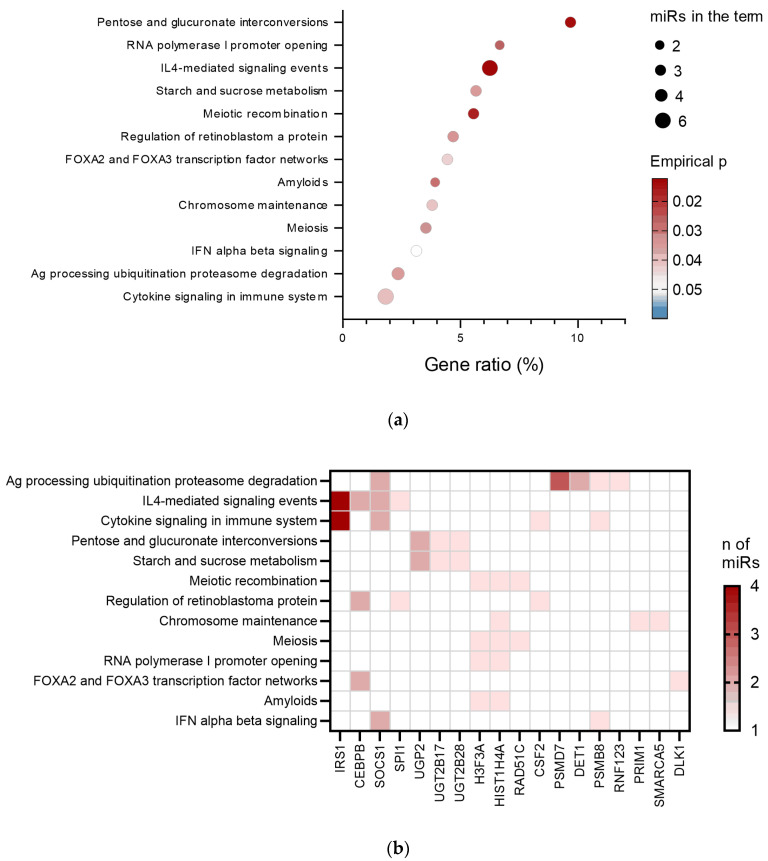
Biological pathways affected by miRNAs differentially expressed in sera of severe-asthma vs. never-asthma patients. (**a**) Bubble chart showing biological pathways affected by identified target genes. X-axis corresponds to the percentage of targets in genes of the identified pathways. The size and the colour of the bubble correspond to the number of miRs of interest involved in a pathway and to enrichment significance, respectively. Pathways are chosen based on the empirical *p*-value cut-off of 0.05. (**b**) Heatmap of target genes in identified pathways (stratified by the percentage of targets in total targets of miRNAs of interest). The colour intensity shows the number of miRs of interest targeting a gene. The analysis was performed using the miRSystem tool.

**Table 1 ijms-26-06676-t001:** Demographic distribution of patients from the three studied groups.

Group	Severe Asthma	Mild Asthma	Never Asthma	*p*-Value
Median age (IQR)	54.0 (40.0–88)	26.0 (25.0–27.0)	18.0 (17.0–18.0)	<0.001
Male, N (%)	15 (30.0)	20 (40.0)	23 (46.0)	<0.001
Female, N (%)	35 (70.0)	30 (60.0)	27 (54.0)	-
Median age at asthma diagnosis (IQR)	28.0 (7.0–67)	6 (0–25)	N.A.	<0.001
Smoking (ever) N (%)	24 (48)	22 (44)	23 (46)	0.91
Median BMI (IQR)	28.9 (24.8–48.3)	25.3 ((17.8–55.5) *	25.1 (18–40.3)	<0.001
Atopy (%)	28 (56.0)	35 (70.0)	16 (32.0)	<0.001
m-OCS N (%)	14 (28)	N.A.	N.A.	-

* BMI data for mild-asthma patients is from their 18-year follow-up. Only parameters available across all three cohorts were included for statistical comparison. Statistical analysis: Kruskal–Wallis test for continuous variables; Chi-square test for categorical variables; significance threshold set at *p* < 0.05. N.A.—Not applicable, IQR—interquartile range, m-OCS— maintenance oral corticosteroids.

## Data Availability

The data and materials used in this study will be made available, as appropriate, on receipt of reasonable request to the study team.

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
