# Peer review of "Circulating microRNAs as Potential Diagnostic Tools for Asthma and for Indicating Severe Asthma Risk"

_ijms, 2025, doi:10.3390/ijms26146676_

Round 1

Reviewer 1 Report

Comments and Suggestions for Authors

The manuscript by Vorobeva and colleagues, "Circulating microRNAs as potential diagnostic tools for asthma and for indicating severe asthma risk", compares microRNA expression in blood samples from severe asthmatics, mild asthmatics, and controls.  The area is clinically important, and this study provides insight into the roles of microRNAs in asthma.  This investigation therefore adds to our understanding of asthma.  However, this study has significant limitations that are outlined below:

  1. Table 1 uses "Na", which is likely an abbreviation for "not applicable".  This abbreviation should be spelled out in the legend, particularly since one of the analysis groups is "never asthma" and so "Na" is confusing.
  2. Figure 1 combines the mild and severe asthmatics into one group.  It would be interesting to have the data presented showing the values for the mild and severe asthmatics as separate groups in comparison to the "never asthma" group.
  3. Figure 2 compares the mild and severe asthmatics.  It would be interesting to include the "never asthmatics" to show how the asthmatics differ from the controls.  Also, the color used for the severe asthmatics is very similar to that used for the "never asthma" group.  The figure would be easier to read if the color used for the severe asthma group was more distinct than that used for the "never asthma" group.
  4. Page 6, line 186 reports that a "considerable proportion of identified pathways were related to inflammation and immune responses".  Since this is in the Results section, it should be more specific and state how many identified pathways were related to inflammation.  If possible, including the total number of identified pathways would give the interested reader a more clear understanding of the abundance of inflammatory pathways modulated by the microRNAs identified in their analysis.
  5. Both of the cohorts used for this study were drawn from the UK.  It would greatly strengthen this analysis to include an analysis of a cohort of asthmatics derived from patients outside of the UK.  

Reviewer 2 Report

Comments and Suggestions for Authors

In the manuscript by Vorobeva et al., authors identified a number of circulating miRNAs deregulated in Asthma patients versus healthy subjects (NA) as well as in severe- versus mild-asthma patients (SA and MA respectively). In particular authors reported that in whole cohort of asthma patients out of the 35 miRNAs analyzed miR223-3p, -191-5p and -197-3p were upregulated whilst miR-155-5p was downregulated compared to NA. Looking at differences between MA and SA patients authors revealed that miR-191-5p and miR-197-3p were increased in MA patients compared to healthy subjects whereas miR-223-3p was upregulated in SA versus MA patients. Further, miR-144-3p, -30a-5p, -660-5p and 125b-5p were all downmodulated in SA versus MA. ROC curves were derived from the data as well as specificity and sensitivity values for each miRNAs as well as for the combination of all the value. Deregulation of some miRNAs was also correlated to clinical parameters currently used to define asthma severity. Finally, looking at the putative affected pathways, authors reported that several identified miRNAs are involved in the regulation of inflammation, cytokine signaling and antigen processing/presentation.

The work done by Vorobeva et al., is well conceived and developed and the results they report sustain the conclusion the authors drawn. I don’t have major criticisms to move to the work just few minor annotations authors should amend:

  1. In the abstract, line 31 the value 76% is reported twice.
  2. In paragraph 2.1 when authors describe Table 1 only some data were reported as statistically significant. Are these the only statistical significant differences between the cohorts? Looking at the data it seems to me that also median age is different. Authors should report statistical analysis for all the data in table 1 or, even better add in the table the p value.
  3. In paragraph 2.2, line 97 authors generically refer the results to fig. 1, but the values they report correspond to fig. 1b. I suggest the authors to refer to fig. 1a at line 92 and at fig. 1b at line 97.
  4. Authors must check the abbreviations and explain them when they use for the 1st time: for ex. FEV1 is used in the results section but it is explained later on in material and methods section.
  5. In Discussion authors should comment the data about specificity and sensitivity obtained from ROC curves. Are 87% sensitivity and 66% specificity for asthma diagnosis panel and 76% sensitivity and 68% specificity for severity risk panel values high enough?
